# Genetic Variability Assessment of Tropical Indica Rice (*Oryza sativa* L.) Seedlings for Drought Stress Tolerance

**DOI:** 10.3390/plants11182332

**Published:** 2022-09-06

**Authors:** Naqeebullah Kakar, Salah H. Jumaa, Saroj Kumar Sah, Edilberto D. Redoña, Marilyn L. Warburton, Kambham R. Reddy

**Affiliations:** 1Department of Plant and Soil Sciences, Mississippi State University, Mississippi State, MS 39762, USA; 2Department of Biochemistry, Molecular Biology, Entomology and Plant Pathology, Mississippi State University, Mississippi State, MS 39762, USA; 3Delta Research and Extension Center, Mississippi State University, 82 Stoneville Road, Stoneville, MS 38776, USA; 4Corn Host Plant Resistance Research Unit, Crop Science Research Laboratory, United States Department of Agriculture—Agriculture Research Service, Mississippi State, MS 39762, USA

**Keywords:** drought, rice, early growth, morpho-physiological parameters, root, shoot, WinRHIZO root image analysis, drought stress, response indices

## Abstract

Drought stress is one of the most devastating abiotic factors limiting plant growth and development. Devising an efficient and rapid screening method at the seedling stage is vital in identifying genotypes best suited under drought conditions. An experiment was conducted to assess 74 rice genotypes for drought tolerance using specially designed mini-hoop structures. Two treatments were imposed on rice seedlings, including 100% moisture and a 50% moisture regime. Several shoot morpho-physiological traits and root traits were measured and analyzed. The genotypes exhibited a wide range of variability for the measured traits, with the leaf area showing the most significant variation, followed by plant height, tiller number, and shoot dry weight. In contrast, the drought did not significantly affect most root traits. The germplasm was classified into different categories using cumulative drought stress response indices (CDSRI); 19 genotypes (26%) were identified as drought sensitive, and 33 (45%), 15 (20%), and 7 (9%) were determined as low, moderately, and highly drought-tolerant, respectively. Genotypes IR86638 and IR49830 were the most and least drought-tolerant, respectively. Overall, a poor correlation was observed between CDSRI, total shoot traits (R^2^ = 0.36), and physiological parameters (R^2^ = 0.10). A strong linear correlation was found between CDSRI and root traits (R^2^ = 0.81), suggesting that root traits are more crucial and better descriptors in screening for drought tolerance. This study can help rice breeders and scientists to accelerate breeding by adopting a mini-hoop rapid screening method. The tolerant genotypes could serve as appropriate donor parents, progenies, and potential genotypes for developing drought-tolerant commercial cultivars.

## 1. Introduction

Rice (*Oryza sativa* L.) is a primary cereal crop, feeding more than half of the world’s population [1]. It is grown as an annual crop in diverse climatic conditions, including tropical, subtropical, semiarid tropical, and temperate regions worldwide. Rice requires adequate water, including standing water or permanent flooding, during its entire lifecycle for optimal growth and development and high yield [2]. Water is the most crucial resource in rice production; however, its limited availability has caused severe drought stress [3]. Drought stress has been ranked as the most devastating abiotic factor in rice-growing areas globally, causing substantial yield reductions [2]. Drought stress is becoming a severe problem due to climate change, projecting a potential risk for rice productivity and food security. It can cause up to 58% yield losses in rice in one growing season [4]. Drought also affects arable land area; the percentage of land affected by drought has doubled from the 1970s to the 2000s, and, unfortunately, the future trend appears to be similar [5]. Drought has been reported to have already affected rice production in most rice-growing areas around the world, including the USA, China, Australia, and many countries in Asia and Africa. In Asia, where more than 90% of the world’s rice is produced and consumed [6], 15 million hectares and 22 million hectares of traditionally irrigated land have been estimated to be suffering from physical water scarcity and economic water scarcity, respectively [7]. Overall, 63.5 million ha of rainfed rice grown annually has been affected by drought, mostly in tropical Asia, Africa, and Latin America [8]. Drought has decreased the world’s overall per capita irrigated area from 48 ha/1000 people in late 1970 to approximately 42 ha/1000 people in 2002 [9].

Drought may occur at any stage in the growing season, including the seedling, flowering, and grain-filling stages. Drought intensity depends on the duration and frequency of water scarcity [10]. The severity of drought also depends upon different environmental factors, including the occurrence and distribution of rainfall, the type of soil and its evaporating requirements, and moisture-storing capacity [11]. The intensity and frequency of drought are expected to increase in the future due to climate change causing changes in temperature, atmospheric carbon dioxide (CO_2_), and the frequency and intensity of extreme weather conditions. These changes result in a soil water deficit, causing a significant threat to sustainable agriculture production for meeting the food demands of the world’s increasing population [12].

Rice is poorly adapted to limited water conditions because of its semi-aquatic nature and origin [13]. It is susceptible to drought stress at the seedling and maturity stages [14,15], limiting rice production and yield stability. The seedling stage is more critical to drought as it is an essential determinant for rice’s later growth and maturation. Therefore, screening for rice cultivars that can tolerate drought stress at the vegetative stage will bring new insights into rice breeding [16].

Although it may not be possible to completely overcome the problem of drought, some agronomic and genetic options could be established to manage crops, including rice, more effectively under water-limited conditions. However, there are currently no economically viable drought tolerance-based mechanisms for enhancing crop production. Therefore, screening for drought stress to identify drought-tolerant genotypes might be a promising approach to sustain crop production under water-limited conditions and ensure food security for the rapidly increasing global population. For efficient screening, detailed physiological and genetic insights into the contributing traits at different plant growth and development stages are imperative [12].

Early vigor and biomass accumulation are valuable, but intricate, attributes for anticipating later reproductive traits in rice. Early vigor refers to the ability of plants to rapidly accumulate biomass and leaf attributes until improved canopy development and closure are attained. It is a promising property involving processes such as resource procurement and conservation, organ and morphogenetic dynamics, and plant development and canopy architecture. Thus, early vigor can contribute to growth, development, and yield stability by the rapid colonization of space and resources [17] under drought conditions. Early canopy closure also helps to reduce unproductive, non-transpirational water use, increasing the overall water use efficiency (WUE) [18], which, in turn, enhances weed competitiveness at an early growth stage [19,20]. Early vigor traits could indicate the fast-growing and short-duration varieties, which can help identify early maturing and high-yielding cultivars. Early maturing cultivars utilize less water per season and offer an escape mechanism to plants to effectively avoid various biotic and abiotic challenges. Thus, earliness is essential for increasing yield and saving time and water consumption.

Root systems play an essential role in plants’ growth and development, planting density, and aboveground biomass. Root attributes have been underestimated and less valued in the past, mainly because of the unavailability of proper scientific instruments to estimate their underground distribution, interactions with the surrounding environment, functional diversity, and complex structures. The WinRHIZO optical scanner and software system is an ideal scientific instrument to study complex root structures and understand their role under stressful conditions. It can measure several root attributes [21], including the number of roots, root surface area, cumulative root length, and average root diameter. Studying root attributes can exploit genetic variability and specific connections with growth, developmental, and yield traits under stressful conditions.

Exploiting genetic variability for drought-related characteristics will help develop drought-tolerant rice genotypes [22,23]. However, little is known about how different traits (especially root traits) respond and express under drought and the trade-offs of crucial attributes for drought tolerance. Therefore, breeders are interested in identifying breeding lines and varieties possessing drought-tolerant genes that could be used in breeding programs as donor parents to improve drought tolerance in high-yielding cultivars [24]. Breeders are also interested in identifying component traits that contribute directly or indirectly to yield and are comparatively easier to measure than the yield itself [25]. Early vigor-related traits used in previous studies have involved root and shoot traits measured at the seedling stage [26].

The current study aimed to explore the morphological characteristics in indica rice genotypes associated with seedling vigor and their manifestation under water-limited conditions. We hypothesized that the variability in morpho-physiological and genetic traits in the selected rice genotypes could be a pre-breeding resource for improving genotypic performance under limited water conditions. The objectives of this study were to: (i) test the efficiency and accuracy of the pot-culture screening method at the early growth stages using mini-hoop structures, (ii) determine the variation in morpho-physiological traits and identify the best descriptors (traits) for drought stress tolerance, (iii) classify and rank rice genotypes into different response groups, and (iv) study the interrelationships among different morpho-physiological traits. We anticipate that the information will allow rice breeders to determine and select drought-tolerant genotypes at the seedling stage. The identified tolerant lines can be used to breed drought tolerance. The short-season pot-culture screening method and early growth stage characterization strategy may be helpful decision-making tools for farmers and crop consultants to select commercial cultivars most suited for water-limited rice production systems.

## 2. Results

### 2.1. Performance of Rice Genotypes and Interaction with Drought

The analysis of variance for shoot growth and developmental parameters revealed significant (*p* > 0.001) differences among the rice genotypes and drought treatments. whereas drought X genotype interactions were non-significant for all traits except leaf area (LA). Traits such as plant height (PH), tiller number (TN), and LA were significantly different among the genotypes under drought treatment (Table 1). Among the root growth and developmental traits, significant differences were found for average root diameter (ARD), root crossings (RC), and root tips (RT) under drought treatment. No significant drought X genotype interaction was observed for any of the root traits.

Significant differences were only found for the treatment’s chlorophyll content and genotypes among the physiological parameters. Still, no considerable drought X genotype interaction was found for any of the physiological traits. All of the aboveground parameters, including leaf dry weight (LDW), stem dry weight (SDW), and shoot dry weight (SHDW), were significantly different under drought treatment and among genotypes. Still, no significant drought X genotype interactions were observed (Table 1).

#### 2.1.1. Shoot Growth and Developmental Parameters

Among the shoot growth and developmental traits, LA was the most diverse and most affected by drought among all genotypes. It ranged from 283.08 cm^2^ per plant in genotype IR86635 to a minimum of 36.64 cm^2^ per plant in genotype IR49830, with an overall average of 119.11 cm^2^ per plant under drought compared to 146.50 cm^2^ under control conditions (Figure 1A). PH ranged from 13.75 cm in genotype MTU1010 to 3.88 cm in genotype IRRI123, with an overall average of 10.23 cm under drought stress compared to the overall average of 12.14 cm under the control conditions. Thus, drought stress reduced PH by 1.91 cm per plant and LA by 27.39 cm^2^ per plant on average (Appendix A). Similarly, TN also decreased significantly under drought treatment, ranging from 3 to 5, and an average of 3.59 tillers per plant under drought.

#### 2.1.2. Root Growth and Developmental Parameters

Most of the major root growth parameters, including LRT, CRL, RSA, and RV, were not significantly affected by drought. However, drought stress affected ARD significantly and increased it from an average of 0.41 in control conditions to 0.43 mm under drought conditions. ARD ranged from 0.36 mm in genotype COL-XXI to 0.48 mm in genotypes IR65600 and IR09L324 (Appendix A). Similarly, RSA ranged from 147.68 cm^2^ in genotype IR49830 to 639.06 cm^2^ in genotype CT18237, with an overall average of 416 cm^2^ under drought compared to 427.20 cm^2^ under control conditions, causing an average reduction of 11.2 cm^2^ per plant (Appendix A).

Only RC revealed significant differences among the root developmental parameters, whereas RT and RF were non-significant variables under drought. RC ranged from 570.75 in genotype IR49830 to 4724.75 in genotype IR86635, with an overall mean of 2623.79 under drought compared to 3089.36 under control conditions. Similarly, RT ranged from 8053.25 in genotype IR49830 to 33,253 in genotype IR86635, with an overall mean of 20,720.57 under drought compared to 22,286.56 under control conditions (Appendix A). Substantial natural variation was observed for the measured root growth and developmental traits under drought (Figure 2). The genotypes responded differently to drought, showing significant variability among the genotypes.

#### 2.1.3. Physiological Parameters

No significant differences were found in most of the physiological parameters (Appendix A) except chlorophyll content (SPAD), which increased from a mean of 39.37 per plant under control to 42.54 under drought, with a range of 32.83 in genotype IR07F102 to 49.43 in genotype CT18593 (Figure 1B). The minimum fluorescence (F_o_) ranged from 9517.75 in genotype IR85422 to 6484.75 in genotype HHZ12, with an average of 8050.27 under drought compared to 7906.45 under control conditions. Similarly, the maximum fluorescence (F_m_) ranged from 24,765 in genotype GMET-25 to 11,184.75 in genotype HHZ12, with an overall mean of 17,958.81 and 17,643.62 under drought and control conditions, respectively. However, the maximum quantum efficiency (F_v_/F_m_) remained unchanged under drought (Appendix A).

#### 2.1.4. Aboveground Biomass

All of the aboveground biomass traits, including LDW, SDW, and SHDW, were reduced significantly under drought. The average LDW declined from 0.91 g in control to 0.71 g under drought, with the highest (1.22 g) and least (0.28 g) LDW found in genotypes IR86635 and IR49830, respectively. SDW ranged from 0.91 g in genotype FED21 to 0.22 g in genotype IR07F102, with an overall mean of 0.58 g under drought stress compared to the overall mean of 0.67 g under control conditions. Similarly, drought also had an adverse effect of 0.29 g on the SHDW per plant. Overall, due to drought, the total aboveground biomass was reduced from 1.58 g in control to 1.29 g (Appendix A). A maximum decline in biomass (2.06 g) was found in genotype IR49830 (Figure 3A), while a minimum reduction (0.57 g) was observed in genotype IR86635 (Figure 3B).

### 2.2. Classification of Rice Genotypes Based on Drought Response Indices

The cumulative drought stress response index (CDSRI) values of all of the measured shoot, root growth and developmental, and physiological traits and their standard deviations were used to classify the rice genotypes into different response groups (Table 2). The 74 rice genotypes were categorized into four response groups; 19 genotypes (26%) were identified as drought-sensitive, 33 (45%) as low drought-tolerant, 15 (20%) as moderate drought-tolerant, and 7 (9%) as highly drought-tolerant genotypes. The lowest CDSRI value (14.44) was found in genotype IRRI 154, which developed a shallow and less extensive root structure and was highly drought-sensitive (Figure 4A). The highest CDSRI value (28.88) was observed in genotype IR86126, maintained extensive root structure, and was highly drought-tolerant (Figure 4B).

Cumulative drought stress response index (CDSRI) values were further used to compute the correlations between the shoot, root, and physiological traits with the cumulative drought stress response index (Figure 5). An overall linear positive correlation was observed between the CDSRI and total shoot traits (R^2^ = 0.36), root traits (R^2^ = 0.81), and physiological parameters (R^2^ = 0.10). A similar strong linear correlation was also observed between the CDSRI and root traits (R^2^ = 0.81) (Figure 5).

## 3. Discussion

Indica rice (*Oryza sativa* L.), or Asian rice, is one of the two main domesticated subspecies of rice [27] and is an essential model for studying crop evolution and the genetic basis of traits in rice [28]. All genotypes used in the current study are indica rice, except for two local checks (Thad and Rex) that belong to the tropical japonica group commonly grown in the US mid-south. The Indica genotypes are primarily breeding lines and have previously not been well characterized for drought stress.

Rice is usually screened for drought in greenhouses under controlled conditions and tested under field conditions in later generations. Both of these screening methods are different and have certain limitations. The greenhouse screening is more accurate because of the controlled conditions, but it is challenging to simulate the natural or field conditions; therefore, cultivars can fail to express similar outputs when transplanted in the field. Similarly, open field-based screening also faces problems such as non-avoidance of rainfall, diseases, and other extreme weather conditions, making it difficult to determine the true potential of cultivars for drought tolerance. Thus, we used “mini-hoop structures”, which have portable canopies that can block rain, but can be removed to provide simulated field conditions on non-rainy days [21]. Therefore, we assume that this alternative screening methodology could be more helpful in assessing and exploring the real phenotypic and genetic variability. It could also be used to screen commercial cultivars under water-limited conditions to identify the best-suited varieties requiring less water for large-scale cultivation.

Previous studies have found that drought stress critically decreases plant growth and development at the vegetative stage in rice [29,30], mainly because of impaired germination [31] and poor seedling stand establishment [32]. According to [33], moisture stress adversely affects plant height at the vegetative stage because of the interruption of cell division and the inhibition of cell elongation under drought stress. Shoot morphological traits, including PH, LA, TN, and LN, are interrelated and can affect each other directly or indirectly. However, the level of diversity and sensitivity varies in different genotypes. Reductions in leaf area, plant height, and overall crop growth are due to impaired mitosis, cell elongation, and expansion under drought [32,34,35]. Drastic reductions in plant developmental traits, including plant height and leaf area, could also be because of the adverse effects of drought on mineral nutrition and metabolism, causing alterations in the assimilate partitioning of plant organs [36].

LA is directly dependent on LN and indirectly reliant on TN. In the current study, we found that drought significantly affected TN. This is a kind of defense mechanism in the plant because plants with higher LA consume more water. In contrast, lower LA in the seedling stage may help plants to survive under limited-water conditions by avoiding additional water losses. Similarly, TN is also one of the main components for higher yield. Breeders try to increase the number of productive tillers and decrease the number of non-productive tillers as they do not play any role in yield enhancement. This would help plants survive drought by avoiding excess water loss through transportation.

Some root growth and developmental traits, including ARD and RV, showed a positive response to drought, indicating that root traits may increase as the plants try to compensate for drought stress. This could occur due to increased soluble sugars in the roots under drought, as sugars provide a source of energy for root development. Soluble sugars are modulated between leaves (source capacity) and roots (sink) by the increased activity of the plant enzymes sucrose phosphate synthase and root invertase [37], which are involved in the plant’s defense response mechanisms to biotic and abiotic stresses. These two enzymes act as energy sources and compatible solutes for osmotic adjustments in roots [38].

The decline in photosynthesis under drought occurs due to a reduction in leaf expansion, photosynthetic machinery, premature leaf senescence, and a related decrease in nutrient production [39]. However, drought can increase chlorophyll content [40] or may have no detrimental effect depending upon the plant species, severity and timing of drought, and growth stage of the crop [41], as well as interactions with other stresses [42]. Water deficit results in decreased stomatal conductance, lower transpiration rate, and lower total water use because of a smaller leaf area index [43]; and depletion of inter-cellular CO_2_ leading to photoinhibition [44].

In the present study, we found that all physiological traits positively responded to drought, particularly chlorophyll content (SPAD), which increased significantly under drought. Minimum and maximum fluorescence (F_o_, F_m_) also increased slightly under drought. Still, this increase was insignificant, showing that chlorophyll fluorescence may not play an essential role in the survival of plants under drought.

Water stress affects plant dry matter by reducing the leaf area, which also slows down the rate of photosynthesis, leading to inadequate assimilation for organ development under drought [5]. Total dry biomass is highly associated with water stress and could be used to estimate drought tolerance [45]. The decline in aboveground biomass was expected, since it results from plant growth and developmental traits, including PH, TN, LN, and LA, which decreased under drought, and all of these traits were correlated.

Vigor response indices have been previously used for the classification and assessment of different crops against various abiotic stresses such as drought tolerance [46], salt tolerance [21], cold tolerance [47], low- or high-temperature tolerance [48,49], and yield traits diversity [50]. The coefficient of determination (R^2^) measures the degree to which a linear regression model fits a dataset. Its value explains the percentage of variation in the output variables defined by the input variables. In the case of the current study, it is the percentage of differences in tolerance index described by each independent variable. A strong linear correlation between CDSRI and root traits (R^2^ = 0.81) indicates a strong correlation or association between the root traits and the drought, showing that root traits are more important in screening for drought-tolerant rice genotypes at the early growth stage.

CDSRI classified the rice genotypes into four different categories, with the majority of the genotypes (48 out of 74, or 65%) identified as possessing low and moderate tolerance to drought, respectively. Correlations between shoot, root, and physiological traits with cumulative drought stress response index revealed the highest positive linear correlation value for root traits, followed by shoot and physiological characteristics. The highest value for roots indicates that root traits play a more critical role in explaining rice genotypes’ total or cumulative drought stress response index. The weak correlations between CDSRI and shoot and physiological traits signify that shoot and physiological traits are inadequate descriptors in screening and selecting rice genotypes for drought tolerance and should not be focused on when screening for drought tolerance at the seedling stage.

## 4. Materials and Methods

### 4.1. Experimental Setup and Germplasm

Seventy-four rice genotypes (Appendix A) were evaluated for response to drought stress. Most of these genotypes (72) belonged to tropical indica rice subspecies obtained from the International Rice Research Institute (IRRI), Los Baños, Philippines, with some local checks (two cultivars) from the US mid-south included for comparison. The experiment was conducted using pre-fabricated mini-hoop structures (Appendix A) at the Rodney Foil Plant Science Research facility of Mississippi State University, Mississippi State, MS, USA (33°28′ N, 88°47′ W), MS, USA. Each structure consisted of a PVC framework with 4 ml polythene wrapping with 2 m width × 1.5 m height × 5 m length.

Seeds were sown in 592 polyvinyl chloride pots (12 cm diameter and 30 cm height) arranged in a randomized complete block design (RCBD) with four replications and 74 rice genotypes each. The pots were filled with the soil medium consisting of 3:1 sand and soil, classified as a sandy loam (87% sand, 2% clay, and 11% silt) with 500 g of gravel at the bottom of each pot. Initially, five seeds were sown in each pot, and the plants were thinned to one plant per pot seven days after emergence. The plants were irrigated three times a day via an automated, computer-controlled drip system with full-strength Hoagland’s nutrient solution [51], delivered at 800, 1200, and 1700 h until drought treatment was imposed. The soil moisture status was monitored using Decagon soil moisture sensors and data loggers (Em-5b data logger), which use capacitance to measure the soil’s water content by measuring the soil’s dielectric permittivity.

### 4.2. Drought Treatments

The imposed treatments included control (C) and drought (D). Drought treatment was set one week after emergence (12 days after sowing) to well-established seedlings where plants received 50% soil moisture throughout the experiment until the final harvest. Soil moisture was recorded and monitored daily until the final harvest through real-time sensors installed within the pots inside the mini-hoop structures. The average relative humidity of about 80 ± 1.2% and net solar radiation availability of approximately 97% under the mini-hoop systems were monitored at various experiment stages using a light meter (Li-250A, LI-COR, Inc., Lincoln, NE, USA). Real-time temperature sensors were used to measure the diurnal temperature regimes where the average day temperature recorded was 35.86 °C, while the night temperatures hovered around 24.78 °C. A similar experiment of screening rice genotypes at the vegetative stage under drought conditions using rainout structures has been previously carried out, revealing the effectiveness of the experimental setup [16].

### 4.3. Measurements

#### 4.3.1. Growth and Developmental Parameters

The shoot growth and developmental parameters, including plant height (PH), tiller number (TN), and leaf number (LN), were measured one day before the final harvest (36 DAS) for all 74 rice genotypes. The leaf area was measured using the leaf-area meter (LI-3100: Li-COR, Lincoln, NE, USA) on the day of harvest before leaf rolling. The leaves and stems were then stored separately in the oven at 75 °C for 5 consecutive days. Other plant components, including leaf dry weight (LDW), stem dry weight (SDW), shoot dry weight (SHDW), and the total dry weights (TDW) of all plants, were measured after a constant dry weight was attained. The growth and developmental parameters were calculated for drought and irrigated conditions for morpho-physiological and genotypic variability comparison among the genotypes.

#### 4.3.2. Measurement of Physiological Parameters

Physiological parameters were recorded two days before harvest (35th DAS) on-site non-destructively using a SPAD meter (SPAD 502, Aurora, IL, USA). SPAD readings for instant chlorophyll measurements were taken at four different positions on the flag leaf of each genotype and then averaged for the final reading. Fluorescence for minimal fluorescence intensity (F_o_), maximal fluorescence intensity (F_m_), maximal variable fluorescence (F_v_), and maximum quantum efficiency or yield (F_v_/F_m_) was measured with the Fluropen 1000 (Photo System Instruments, Kolackova, Czech Republic) for chlorophyll fluorescence fast-transient (OJIP) analysis to monitor chloroplast function [52] and Photosystem II efficiency. This will give us clues about the stress effect on the experimental rice lines.

#### 4.3.3. Root Image Acquisition and Analysis

At the final harvest, the roots of all plants were cut from the stems and washed on a sieve thoroughly, but cautiously, to avoid any destruction to the overall root structure. The longest root length (LRL) was measured using a metric ruler, root crowns were cut and cleaned, and individual root structures were scanned using the WinRHIZO optical scanner (Regent Instruments, Inc., Québec, QC, Canada). First, the 0.3 by 0.2 m Plexiglas tray was filled with approximately 5 mm of tap water, ensuring that the roots floated in the tray and were easily untangled and separated with a plastic paintbrush to minimize overlapping. The tray was placed on top of a specialized dual-scan optical scanner linked to a computer system. Root images were acquired by setting the parameters to high resolution (800 by 800 dpi) as reported and described previously [40,41]. The acquired images were analyzed for different root parameters, including root surface area (RSA), cumulative root length (CRL), average root diameter (ARD), root volume (RV), number of root tips (RT), number of root forks (RF), and number of root crossings (RC).

### 4.4. Data Analysis

The experiment was laid out using a randomized complete block design that considered the rice lines and drought treatments as the primary sources of variation. Data from all measurements of the root and shoot parameters were documented, and descriptive analysis, including means, standard deviations (SD), coefficients of variation (CV), and analysis of variance (ANOVA), were calculated for the parameters under drought and control treatments using the SAS statistical software packages (SAS Institute, Inc., Cary, NC, USA). The data were analyzed using a one-way ANOVA via PROC GLM in SAS to determine the effect of drought on the shoot and root growth and developmental and physiological parameters. The Fisher’s protected least significant difference test at *p* = 0.05 was employed to test differences among the treatments for the measured parameters. The standard errors of the mean were calculated using Sigma Plot 13.0 (Systat Software, Inc., San Jose, CA, USA) and presented in the figures as error bars.

### 4.5. Drought Response Characterization

The selected rice genotypes were classified into different clusters or response reaction groups based on their responses to drought stress and subsequent summation of individual index values for each trait, as previously explained by [53]. The combined drought stress response indices (CDSRI) were calculated by adding individual drought stress response indices (IDSRI) for all parameters. Initially, IDSRI values for each parameter were calculated as the value of a parameter under drought (P*d*) for a given rice genotype divided by the value for the same parameter under controlled conditions (P*c*) of the same cultivar, as follows:

IDSRI = P*d* /P*c*; and
CDSRI = (PH*d*/PH*c*) + (TN*d*/TN*c*) + (LA*d*/LA*c*) + (LW*d*/LW*c*) + (SW*d*/SW*c*) + (RW*d*/RW*c*) + (TW*d*/TW*c*) + (LRL*d*/LRL*c*) + (F0*d*/F0*c*) + (FM*d*/FM*c*) + (FV*d*/FV*c*) + (F_v_/F_m_*d*/F_v_/F_m_*c*) + (TRL*d*/TRL*c*) + (SA*d*/SA*c*) + (AD*d*/AD*c*) + (RV*d*/RV*c*) + (RN*d*/RN*c*) + (TP*d*/TP*c*) + (FR *d*/FR*c*) + (CR*d*/CR*c*).

Based on the CDSRI values, the rice genotypes were classified into four response groups: drought-sensitive, low drought-tolerant, moderately drought-tolerant, and highly drought-tolerant.

## 5. Conclusions

Drought stress is a highly destructive and damaging abiotic factor limiting plant growth and development in rice-growing areas worldwide. Therefore, devising an efficient and rapid screening technique at the seedling stage is vital in identifying genotypes best suited for limited water conditions. This study was primarily conducted for screening rice genotypes (mainly from the indica rice subspecies) for drought tolerance using specially designed mini-hoop structures to avoid rainfall. Overall, the studied rice genotypes exhibited substantial variability in all of the measured shoot, root, and physiological traits in response to drought stress. Drought caused a significant 20%, 19%, and 16% decrease in LA, TN, and PH, respectively, within 25 days of treatment imposition at the early growth stage, and all aboveground biomass parameters were significantly reduced under drought. The drought had no significant effect on the physiological traits except chlorophyll content (SPAD) and no significant impact on the root growth and developmental traits except ARD. A strong correlation between CDSRI and root measurements indicates that root traits could be more crucial in screening and selecting rice genotypes for drought tolerance at the early growth stage. Most of the genotypes (65%) used in the current study revealed low to moderate drought tolerance, but 9% exhibited high tolerance to drought stress. These genotypes may be helpful for breeders as potential parents of drought-tolerant high-yielding rice cultivars for future commercial production. They can also be used to understand the mechanisms underlying drought tolerance, which has become increasingly necessary for rice breeders to achieve and sustain global food security.

## Figures and Tables

**Figure 1 plants-11-02332-f001:**
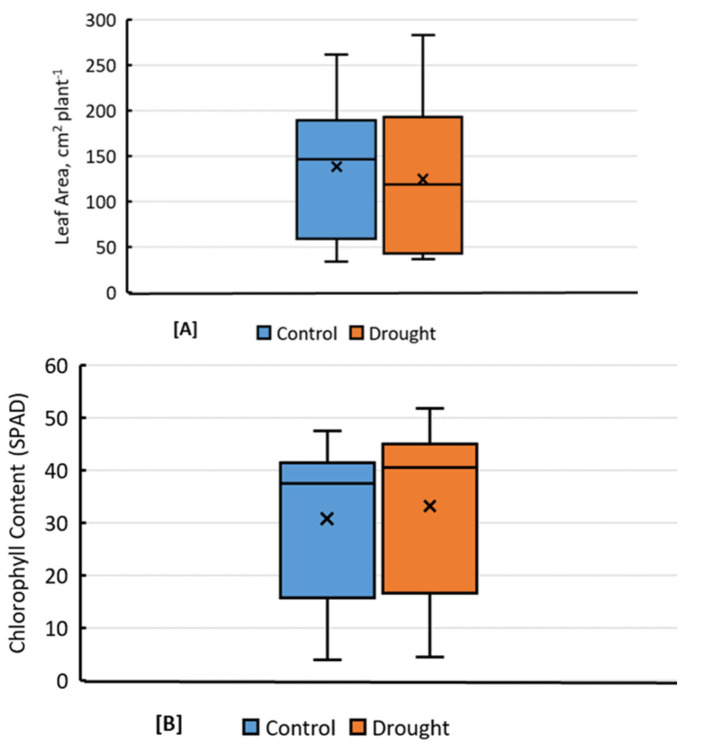
(**A**,**B**) Comparison of 74 rice genotypes for (**A**) leaf area and (**B**) chlorophyll content (SPAD units) traits under drought and controlled conditions observed 37 days after sowing.

**Figure 2 plants-11-02332-f002:**
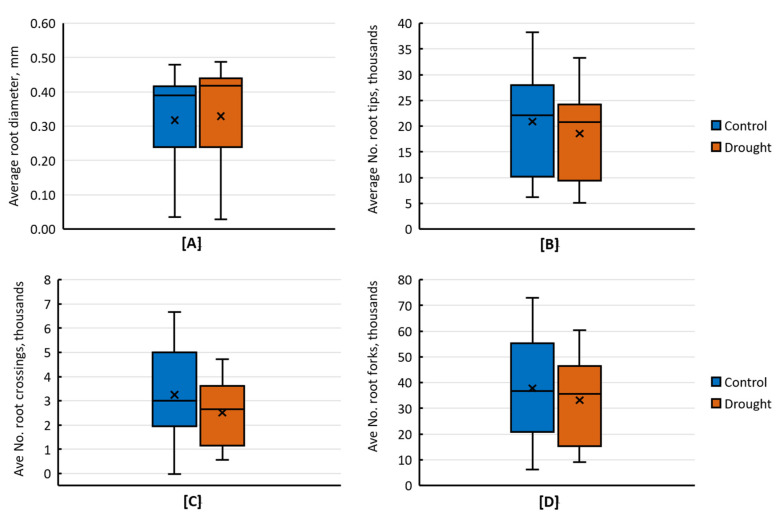
Box and whisker plots for root growth and developmental traits showing natural variation and the effect of drought treatment on the average (**A**) root diameter, mm plant^−1^, (**B**) number of root crossings, (**C**) number of root tips, and (**D**) number of root forks. The whisker below the box represents the first quartile (Q1) or the fifth percentile showing the first 25% data distribution in this range. In contrast, the whisker above the box represents the third quartile (Q3) or 95th percentile showing the last 25% of the data distribution. The length of the box is called the interquartile range (IQR) or (25th to 75th percentile), shows 25% to 75% of the data distribution for that particular trait, and the horizontal line in the box indicates the median value.

**Figure 3 plants-11-02332-f003:**
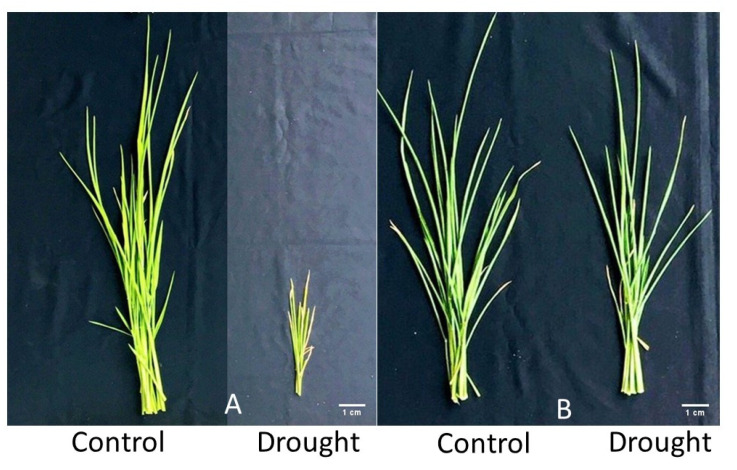
(**A**) Effect of drought on the biomass of drought-sensitive (IR49830) and (**B**) drought-tolerant (IR86635) rice genotypes showing maximum (2.06 g) and minimum (0.57 g) declines of the biomass observed 37 days after sowing. Scale bar = 1 cm.

**Figure 4 plants-11-02332-f004:**
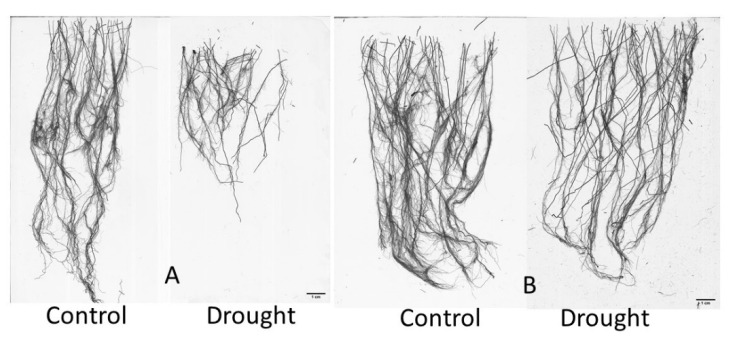
(**A**) Effect of drought on the root structures of drought-sensitive (IRRI 154) and (**B**) drought-tolerant (IR86126) rice genotypes; the drought-tolerant genotype shows more extensive root structure than the drought-sensitive rice genotype, analyzed after harvest using a WinRHIZO optical scanner and software system. Scale bar = 1 cm.

**Figure 5 plants-11-02332-f005:**
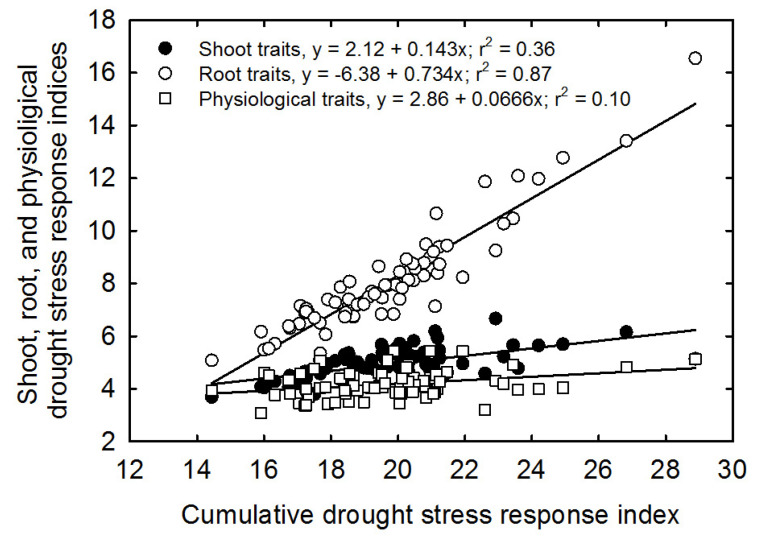
Relationship of the cumulative drought stress response index with the cumulative shoot, cumulative root, and cumulative physiological stress response indices for all 74 rice genotypes. Measurements were taken before harvest (37 DAS) for the shoot and physiological traits and after harvest for root traits.

**Table 1 plants-11-02332-t001:** Analysis of variance across the 74 tropical rice genotypes (G), drought treatment (T), and their interaction (T*G) for the morpho-physiological traits measured at the final harvest, 36–37 days after sowing.

Source	PH	TN	LA	LRL	CRL	RSA	ARD	RV	RT	RF	RC	SPAD	FO	FM	F_v_/F_m_	LDW	SDW	ShDW
Treatment (T)	***	***	***	NS	NS	NS	***	NS	*	NS	**	***	NS	NS	NS	***	***	***
Genotypes (G)	***	NS	***	***	***	***	***	***	**	***	***	***	**	**	*	***	***	***
T*G	NS	NS	**	NS	NS	NS	NS	NS	NS	NS	NS	NS	NS	NS	NS	NS	NS	NS

Significance level ***, **, * and NS means *p*-value ˂ 0.001, 0.01, 0.05, and not significant, respectively. Plant height (PH), tiller number (TN), leaf area (LA), longest root length (LRL), cumulative root length (CRL), root surface area (RSA), average root diameter (ARD), root volume (RV), number of root tips (RT), number of root forks (RF), number of root crossings (RC), SPAD, minimal fluorescence intensity (F_o_), maximal fluorescence intensity (F_m_), the quantum efficiency of fluorescence (F_v_/F_m_), leaf dry weight (LDW), stem dry weight (SDW), shoot dry weight (ShDW).

**Table 2 plants-11-02332-t002:** Classification of 74 tropical rice genotypes into different groups, including drought-susceptible (DS), low drought-tolerant (LDT), moderately drought-tolerant (MDT), and highly drought-tolerant (HDT) genotypes, using shoot and root combined drought response indices of morpho-physiological parameters at the seedling stage.

DS		LDT		MDT		HDT	
Genotypes	CDSRI	Genotypes	CDSRI	Genotypes	CCDSRI	Genotypes	TDSRI
IRRI 154	14.439	HHZ 12	18.281	IR09A130	20.776	IR78222	23.16
IRRI 157	15.91	IR78221	18.414	IR08A172	20.784	CT18233	23.443
IR06N155	16.005	IR75483	18.439	IR09L337	20.835	IR07K142	23.592
PALMAR	16.141	FED21	18.523	IR05F102	20.965	CT18247	24.203
IR86-11	16.321	IR05N412	18.532	GMET-15	20.966	CT18372	24.93
IR04A115	16.741	IRRI 152	18.554	IR86052	21.066	IR86635	26.825
CT18593	16.773	IR64	18.673	IR09L179	21.115	IR86126	28.884
BR47	17.051	IR09N537	18.674	IR86-44	21.147		
IR07F102	17.089	IR6	18.784	CT6946	21.184		
IR86-1	17.223	IR85411	18.977	COL-XXI	21.235		
MIL240	17.253	FED-MO	19.12	Rex	21.245		
IR85427	17.263	CT18245	19.218	CT18237	21.47		
IR08N136	17.319	IR09L324	19.297	HHZ 1	21.938		
Thad	17.505	CT18244	19.423	IR70213	22.603		
CT18615	17.668	FED473	19.51	CT19561	22.92		
IR85422	17.678	FED20	19.529				
IR10A134	17.841	MTU1010	19.617				
Apo	17.904	IR88633	19.688				
IR74371	18.127	GMET-25	19.872				
		IR93324	19.937				
		IR10N230	20.052				
		IRRI 123	20.054				
		IR65600	20.066				
		IR49830	20.093				
		WAB 56	20.122				
		75-1-127	20.161				
		CT18614	20.223				
		IR93323	20.25				
		IR07F287	20.303				
		IR78049	20.312				
		IR65482	20.444				
		CT6510	20.469				
		IR09F436	20.511				
No. of genotypes: 19		33		15		7	
Percentage: 26 %		45%		20%		9%	

## Data Availability

The data are contained within the manuscript. All data, tables, and figures in the manuscript are original.

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
