# Peer review of "Genetic Variability Assessment of Tropical Indica Rice (Oryza sativa L.) Seedlings for Drought Stress Tolerance"

_plants, 2022, doi:10.3390/plants11182332_

Round 1

Reviewer 1 Report

Comments: This manuscript is generally well written except the errors identified below that require major revision.

Suggestions for mandatory changes:

1.       Provide fully spelled-out terms before giving abbreviated terms in parentheses in the legends of tables, figures, as well as the text, because Materials and Methods section appeared later then the Results section.

2.       Clarify “chlorophyll content (SPAD)” by describing what is SPAD meter and how does it work.

3.       Figures 3 and 4 should have the same plant materials representing drought sensitive and drought-tolerant genotype, with the figure 3 showing the shoot biomass and the figure 4 for root biomass. Furthermore, figure 3 is not in agreement with the text on lines 219 to 221. Figure 4 is mentioned before Figure 3. Corrections are needed.

4.       Figure 4 does not demonstrate drought effect (drought vs. control) in two genotypes, which should be same as in the figure 3 - drought-sensitive IR49830 and drought-tolerant IR86635.

5.       If drought had no significant effect on the root growth and developmental traits except average root diameter (ARD), how could a strong correlation between CDSRI and root measurements exist to indicate that root traits could play a more crucial role in screening and selecting rice genotypes for drought tolerance at the early growth stage. Please explain or elaborate.

Author Response

Dear Reviewer

We are glad to receive your valuable suggestions for our manuscript. Thank you very much for your kind consideration on this manuscript “Genetic Variablitiy Assessment of Tropical Indica Rice (Oryza sativa L.) Seedlings for Drought Stress Tolerance”

We are thankful to you  for pointing out several modifications. This manuscript has been amended according to the opinions, suggestions, and comments and all the changes have incorporated in the text using track changes options. The response to all the comments and suggestions has been attached here.

Reviewer 2 Report

The manuscript by Kakar et al. assesses the genetic variability in tropical Indica rice seedlings with respect to drought tolerance. The authors have accomplished this through analyzing shoot morpho-physiological and root traits. Based on these analyses, they categorized the genotypes into different groups for drought tolerance. 

Over all, study is well designed and appropriately executed. However I have a few concerns:

1. My main concern is that if the genotypes under drought showed highest variation in leaf traits (leaf area) and the lowest in root traits (root traits not affected among the genotypes under drought), then why was a poor correlation observed between CDSRI and total shoot traits (R2 = 0.36) and physiological parameters (R2 = 0.10), and a strong linear correlation was found between CDSRI and root traits? In other words, when root traits among rice genotypes didn't vary at all under drought, how could authors correlate these traits with the differential drought tolerance. A clear cut answer needs to be provided and things should be well discussed in the DISCUSSION section.

2. It would be even better if authors could correlate this morph-physiological trait variation among rice genotypes under drought with some drought-responsive molecular/genetic markers.

3.   There are several minor mistakes in writing. I am mentioning here only a few, but authors should check the entire manuscript for such mistakes and correct them. 

a) Line 121: what is morph-genetic? You haven't done any genetic variability study!

b) Line 276: should be "According to,...

c) Line 287: delete the gap b/w "drought" and "which".

d) Line 314: delete the extra ".". Make o and m as subscript in Fo and Fm.

e) Line 367: close the bracket ")".

Author Response

Dear Reviewer

We are glad to receive your valuable suggestions for our manuscript. Thank you very much for your kind consideration on this manuscript “Genetic Variablitiy Assessment of Tropical Indica Rice (Oryza sativa L.) Seedlings for Drought Stress Tolerance”

We are thankful to you for pointing out several modifications. This manuscript has been amended according to the opinions, suggestions, and comments and all the changes have incorporated in the text using track changes options. The response to all the comments and suggestions are attached here.

Reviewer 3 Report

In the manuscript by Kakar et al. entitled "Genetic Variability Assessment of Tropical Indica Rice (Oryza sativa L.) Seedlings for Drought Stress Tolerance", the authors are mainly addressing the variability of drought response by 74 rice lines for the shoot and root traits which led to characterize and categorize the studied genotypes based on mini-hoop screening method.   

This area of research is interesting and scientifically sound good. The authors have written the manuscript very well. The manuscript is presented in a very simple form. The introduction is informative and sufficient. Methods and results are well described and the discussion section is justified with the obtained findings and with valid citation of relevant literature. But this article is not without its few drawbacks, which I am describing below:

Keywords: Please change the long keywords to more than 2 words. For example, 'cumulative drought stress response index' is too long.

Line 43-47, 101-104, 111-114: Add reference.

Table 1: Please mention the full form of all used traits' abbreviations as a footnote. Also, please make the 'Drought (D)' as 'Treatment (T)' and 'D*L' will be 'T*L'.

Figure 1: Please change it to only a simple boxplot with SE bar for both Control and Drought.

Figure 2: Please the title for Y-axis, instead of writing above each graph.

Figures 3 and 4: Add a scale bar.

Figure 5: Please mention the significant level for each trait line.

Line 347: It would be better if you mention the number like '(#72)'.

Line 385: Why so long time (25 days) for drying the plant material at 75ºC? I think 3-4 days are enough at 60-65 ºC.

Based on these points of major needed corrections, I would like to assign 'major revision' of this manuscript in its present form. The author should correct the all changes and submit the final version for further evaluation to be considered for publication. 

Author Response

(The authors gave the same response as above.)

Round 2

Reviewer 1 Report

Table 1: The variables should be Genotypes (G), Treatments (T; control vs drought), and G X T interaction; not Drought (D), Treatment/Lines (T), and D * T. 

Author Response

Dear Reviewer and Editor,
We appreciate the suggestions and comments. The manuscript has been improved and reads much better now.
We thank you again for your constructive review.
We want to state that one of the authors is a former Editor-in-Chief of Crop Science and read the manuscript thoroughly before submitting it to the journal.
We made all the suggestions made by the reviewer, and we hope it is acceptable for publication consideration.
Best,
KR Reddy

Reviewer 3 Report

The authors have corrected all the changes and now it can be accepted for publication. 

Author Response

Dear Reviewer,

We thank you for your input.

Best,

KRReddy